# Core-Shell Hierarchical Fe/Cu Bimetallic Fenton Catalyst with Improved Adsorption and Catalytic Performance for Congo Red Degradation

**Haimei Chen** [1]**, Shaofei Wang** [2]**, Lilan Huang** [1,*] **, Leitao Zhang** [3]**, Jin Han** [1]**, Wanzheng Ren** [1]**, Jian Pan** [1,*]
**and Jiao Li** [1,*]

1    School of Material Science and Engineering, Shandong University of Technology, Zibo 255049, China
2    College of Chemistry and Chemical Engineering, Hunan University, Changsha 410082, China
3    School of Chemical Engineering and Pharmaceutics, Henan University of Science and Technology, Luoyang 471000, China
*    Correspondence: huanglilan_sdut@163.com (L.H.); panjian@sdut.edu.cn (J.P.); haiyan9943@163.com (J.L.)

**Abstract:** The preparation of heterogeneous Fenton catalysts with both adsorption and catalytic properties has become an effective strategy for the treatment of refractory organic wastewater. In this work, 4A-Fe@Cu bimetallic Fenton catalysts with a three-dimensional core-shell structure were prepared by a simple, template-free, and surfactant-free methodology and used in the adsorption and degradation of Congo red (CR). The results showed that the open three-dimensional network structure and the positive charge of the surface of the 4A-Fe@Cu catalyst could endow a high adsorption capacity for CR, reaching 432.9 mg/g. The good adsorption property of 4A-Fe@Cu for CR not only did not inactivate the catalytic site on 4A-Fe@Cu but also could promote the contact between CR and the active sites on the catalyst surface and accelerate the degradation process. The 4A-Fe@Cu bimetallic catalyst exhibited higher catalytic activity than monometallic 4A@Cu and/or 4A-Fe catalysts due to low work function value. The effects of different pH, $H_2O_2$ dosages, and catalyst dosages on the catalytic performance of 4A-Fe@Cu were explored. In the conditions of 7.2 mM $H_2O_2$, 2 g/L 4A-Fe@Cu, and 1 g/L CR solution, the degradation ratio of CR by 4A-Fe@Cu could reach 99.2% at pH 8. This strategy provided guidance to the design of high-performance Fenton-like catalysts with both adsorption and catalysis properties for dye wastewater treatment.

**Keywords:** fenton reaction; Fe/Cu bimetallic catalyst; congo red; adsorption; catalysis

## 1. Introduction

With the advancement and development of science and technology, the discharge of printing and dyeing wastewater is increasing year by year [1,2]. Printing and dyeing wastewater has the characteristics of high chroma, and the dyes contain groups such as benzene, naphthalene, and anthraquinone. [3], which are difficult to be degraded naturally. Some dyes can cause cancer, posing a great threat to human health and safety. Therefore, the treatment of dye wastewater has become an urgent issue to be solved [4,5].

Common treatment methods of dye wastewater include the physical method, chemical method, and biological method [6]. However, for water with relatively high concentrations of pollutants, it is difficult to meet the national emission standards by using traditional methods [7]. In recent years, Fenton/Fenton-like system, as an advanced oxidized water treatment technology, has quickly become a research hotspot due to its high reaction rate, low toxicity, and easy operations [8–14]. The Fenton reaction produces $^\bullet OH$ with a strong oxidizing property through the reaction of ferrous ion (II) with $H_2O_2$, and organic pollutants can be oxidized and degraded into small molecules such as $CO_2$ and $H_2O$ [15–17]. However, the Fenton process has typically been considered to be a highly effective reaction at acidic conditions (pH 2–3) since the iron aqua complexes are insoluble at higher pH values [18,19]

and rather weak above pH 4 [20–22]. Other problems such as loss of the catalyst, leaching of ferric ion, low utilization efficiency of $H_2O_2$, and the accumulation of iron precipitates [23] restricted the environment-friendly use of the Fenton catalyst [24,25]. To solve these drawbacks, many researchers use solid substrates such as inorganic materials [26–28] or polymers [29,30] as the support for immobilizing iron or ferrous ion to obtain heterogeneous Fenton or Fenton-like catalysts. Among the numerous solid matrixes, zeolites are widely used as catalyst supports to immobilize ferric ions due to their uniform, small pore size, large internal surface area, flexible frameworks, and controlled chemistry [31]. Our group used 4A zeolite as the matrix and prepared isolated iron (4A-Fe) Fenton-like catalyst via lyophilization, which exhibited higher catalytic activity and utilization efficiency of $H_2O_2$ in phenol degradation [32]. However, the pH range of the Fenton reaction was still narrow (pH 2–3), and the catalytic efficiency of 4A-Fe reduced rapidly with the pH increased to above 4. The introduction of copper to the catalyst can overcome the difficulty and broaden the pH range of the Fenton process because the copper complex $[Cu(H_2O)_6]^{2+}$ is soluble at neutral pH conditions [33–35], and Cu(I) could be quickly oxidized to Cu(II) on the timescale of minutes at circumneutral pH [36,37]. At present, a lot of Fe/Cu bimetallic catalysts have been prepared and realized the Fenton catalysis at a wide range of pH. However, most Fe/Cu bimetallic catalysts exhibited reduced catalytic efficiency due to their solid construction, which leads to the synergy of Fe and Cu being insufficient [38,39]. The well-defined core-shell hierarchical and open network nanostructures of the bimetallic catalyst may improve the efficiency of catalysts.

In recent years, a catalyst with both adsorption and catalytic properties has become a research hotspot. Nie et al. [40] have prepared a novel hexapod-like pyrite nanosheet mineral cluster catalyst by simple hydrothermal methods. This catalyst could effectively remove ciprofloxacin (CIP) through adsorption and heterogeneous Fenton catalytic reaction, and it has higher catalytic activity and pH range of application than classical homogeneous. Li et al. [41] fabricated the CNTs-CoFe$_2$O$_4$@PPy magnetic nanohybrid and then used it as an adsorbent and catalyst to remove anionic and cationic dyes, the maximum adsorption capacities of the composite materials for anionic dyes methyl blue (MB), methyl orange (MO), and acid fuchsin (AF) were 137.0, 116.1, and 132.2 mg/g, respectively. When it was used as a catalyst to activate peroxymonosulfate (PMS) for the degradation of methylene blue (MEB, 100 mg/L), the 95% of degradation ratio was reached after 1 h. Duan et al. [42] synthesized magnetic activated carbon (MAC) from activated carbon (AC) by impregnation of iron oxides in the porous structure. The porosity of MAC provides abundant adsorption sites for 4-chlorophenol (4-CP), and the adsorption capacity could reach 128.2 mg/g. When it acted as a heterogeneous Fenton catalyst, the degradation ratio of 4-CP reached 92%. Lahiri et al. [43] synthesized a novel nanoporous NiO@SiO$_2$ photo-catalyst by a simple ion-exchange method to degrade the dye, the catalyst exhibits excellent adsorption performance due to its large specific surface area and abundant electron-withdrawing groups, and the catalyst also displayed high catalytic activity over a wide range of pH (3–9). These studies showed that both catalysis and adsorption can effectively remove refractory pollutants in wastewater. However, the relationship between adsorption and catalysis was not demonstrated.

Herein, a simple and template-free method was used to fabricate a core-shell hierarchical 4A-Fe@Cu bimetallic catalyst to solve the above drawbacks. The 4A-Fe served as core components, and two-dimensional copper hydroxide nanosheet assemblies with open network structure were grown in situ on 4A-Fe and served as the shell. The 4A-Fe@Cu catalyst was characterized, and the adsorption and catalysis of anionic dyes (Congo red, CR) were investigated. As a comparison, the adsorption of cationic dyes (Rhodamine B, RhB) by 4A-Fe@Cu was also conducted at the same conditions. In addition, the relationship between adsorption and catalysis was studied intensively and the mechanism of the whole reaction was attempted to propose.

## 2. Results and Discussion

### 2.1. Catalyst Characterization

The size and morphology of as-prepared catalysts are shown in Figure 1. Three different types of catalysts with particle sizes are around 1–3 μm. Compared with the smooth surface of 4A zeolite, the surface of 4A-Fe is rough with iron species loading. It has been certified that the iron species is located inside the 4A zeolite skeleton in the form of $[FeO_4]$ [32]. After the in situ growth of copper-containing nanosheets on the surface of 4A zeolite, the 4A@Cu catalysts were obtained. The SEM (Figure 1c) shows that two-dimensional copper-containing nanosheets were uniformly distributed on the surface of 4A zeolite. When 4A zeolite was replaced by 4A-Fe, the 4A-Fe@Cu Fenton-like catalyst was synthesized (Figure 1d–i). It can be obviously observed that the pH of the $CuCl_2$ solution has a great influence on the surface structure of the catalyst. When the pH of $CuCl_2$ is lower than 1.4, there are no nanosheets structure on the surface obtained 4A-Fe@Cu. This is mainly because there are too many growth sites, leading to the copper sheets all being laid flat on the surface of 4A-Fe, and they are linked together to form a whole. As the pH of the $CuCl_2$ solution increases, the open network structure constructed by interconnected nanosheets on the 4A-Fe@Cu surface is more obvious. When the pH of the $CuCl_2$ solution is 1.6, the perfect network structure constructed by thousands of interconnected nanosheets is observed. The pore size constructed by nanosheets is $0.1-0.2$ μm. This structure ensures that the catalytic sites of the iron catalyst fixed in 4A-Fe@Cu are not embedded; thus, the catalytic activity is not restrained. In addition, the three-dimensional network structure can endow the catalyst adsorption performance. As the pH of $CuCl_2$ further increased, the copper-containing nanosheets began thickening, and particulate matter appeared. When the pH reaches 3.0, the network structure on the catalyst disappears. This is mainly because the copper-containing sheets are too thick in this condition.

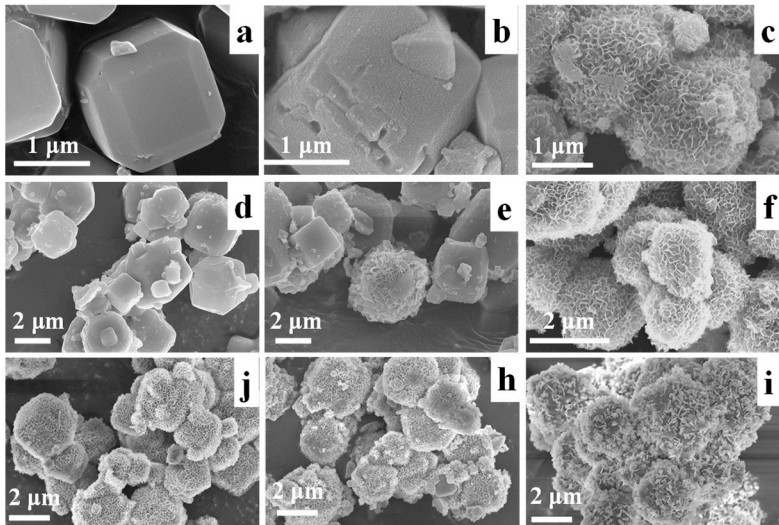

**Figure 1.** The SEM images of 4A zeolite (**a**), 4A-Fe (**b**), 4A@Cu (**c**), and 4A-Fe@Cu ((**d**–**i**), pH of $CuCl_2$ is 1.2 (**d**), 1.4 (**e**), 1.6 (**f**), 1.8 (**g**), 2.0 (**h**), and 3.0 (**i**)).

From TEM analysis (Figure 2a–f), a three-dimensional core-shell structure with 4A-Fe as core and two-dimensional copper-containing nanosheets assemblies as the shell can be clearly observed, and the thickness of nanosheets is about 150–250 nm. This SAED pattern reveals that the nanosheet structure on the 4A-Fe@Cu surface is amorphous. The core-shell hierarchical structure feature of the 4A-Fe@Cu catalyst was further confirmed by EDS mapping. The Fe element is evenly distributed in the 4A zeolite framework (core region) (Figure 2d), and the mass content is about 2.61 wt.% (Figure 2f), while the Cu element distributes in the whole area of the particle (Figure 2e), and the mass content is 49.79 wt.% (Figure 2f).

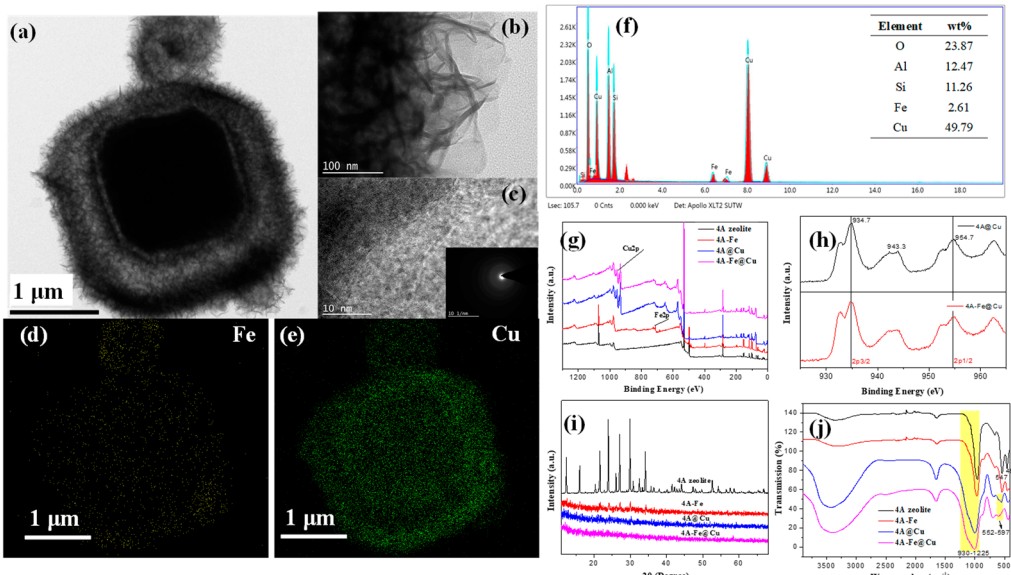

**Figure 2.** (**a**–**c**) TEM images of 4A-Fe@Cu and the inset is the corresponding SAED pattern, (**d**–**f**) is the corresponding EDS mapping images of Fe and Cu and EDS energy spectrum, (**g**,**h**) XPS wide scan and Cu2p core-level spectra of catalysts, (**i**) XRD patterns and FTIR spectra (**j**) of catalysts.

The XPS spectra of catalysts are shown in Figure 2g,h. There is a wider shake-up satellite at about 8.4 eV higher than the $Cu2p_{3/2}$ main peak located at 943.3 eV (Figure 2h). The facts rule out the possibility of the presence of mono-valence copper ions and provide solid evidence that CuO and $Cu(OH)_2$ phase co-exist in 4A@Cu and 4A-Fe@Cu [44]. The XRD patterns of as-prepared catalysts are shown in Figure 2i. The zeolite exhibits a good crystalline structure with sharp diffraction peaks. After loading Fe and Cu, the sharp diffraction peaks completely disappear, indicating the amorphization and decreased specific surface of catalysts [45,46]. The FTIR spectrum of catalysts is shown in Figure 2j. The vibration bands in the 547 cm$^{-1}$ and 462 cm$^{-1}$ weaken or disappear in iron- and/or copper-containing samples compared with zeolite, suggesting that most of the double-four-ring (D4R) units that represent the Linde Type A (LTA) framework [47] are destroyed. This illustrates that zeolite underwent irreversible amorphization, confirmed by the XRD analysis above. The peaks in the region of 930 cm$^{-1}$–1250 cm$^{-1}$ indicate the asymmetric stretching vibrations of the internal [TO$_4$] (T = Si, Al) tetrahedral still existed [47]. In addition, for copper-containing catalysts, a band at 552–597 cm$^{-1}$ is the characteristic peak of the Cu-O stretching vibration from copper hydroxide [48,49].

### 2.2. Adsorption Performance

The adsorption capacities of various catalysts for CR are shown in Figure 3. CR can hardly be adsorbed by 4A zeolite and 4A-Fe catalysts. 4A@Cu and 4A-Fe@Cu have excellent adsorption properties for CR dye. In particular, the maximum adsorption capacity of the 4A-Fe@Cu catalyst for CR reaches 432.9 mg/g, which is higher than that of 4A@Cu (366.8 mg/g). This may be ascribed to the positive charge density on catalysts. The surface potential of the catalyst was conducted by KPFM, and the results are shown in Figure 4. The catalysts are positively charged, and 4A-Fe@Cu possesses the highest surface potential of 37.31 mV, which indicates a high electron overflow capacity and charge density. To confirm this speculation, the adsorption of cationic dyes (Rhodamine B, RhB) by 4A-Fe@Cu was carried out, and the result (Figure 3b) shows that RhB adsorption capacity is very low due to the repulsion between the same charges. In addition, as shown in Figure 3c, the 4A-Fe@Cu without an open network structure exhibits low CR adsorption capacity. As the network structure increases, the CR adsorption capacity increases. This is mainly because the adsorption sites increase as the copper-containing nanosheet structure forms on the catalyst surface. The above results suggest that the adsorption of CR on 4A-Fe@Cu

is attributed to an open three-dimensional network structure and the positive charge of 4A-Fe@Cu. The adsorption of CR by 4A-Fe@Cu was further visualized by SEM; the results are shown in Figure 3d,e. From the comparison, it can be clearly observed that the pores on the catalyst surface constructed by copper-containing nanosheets are almost filled by CR molecules and the color of 4A-Fe@Cu changes from yellow-green to red-brown after adsorption.

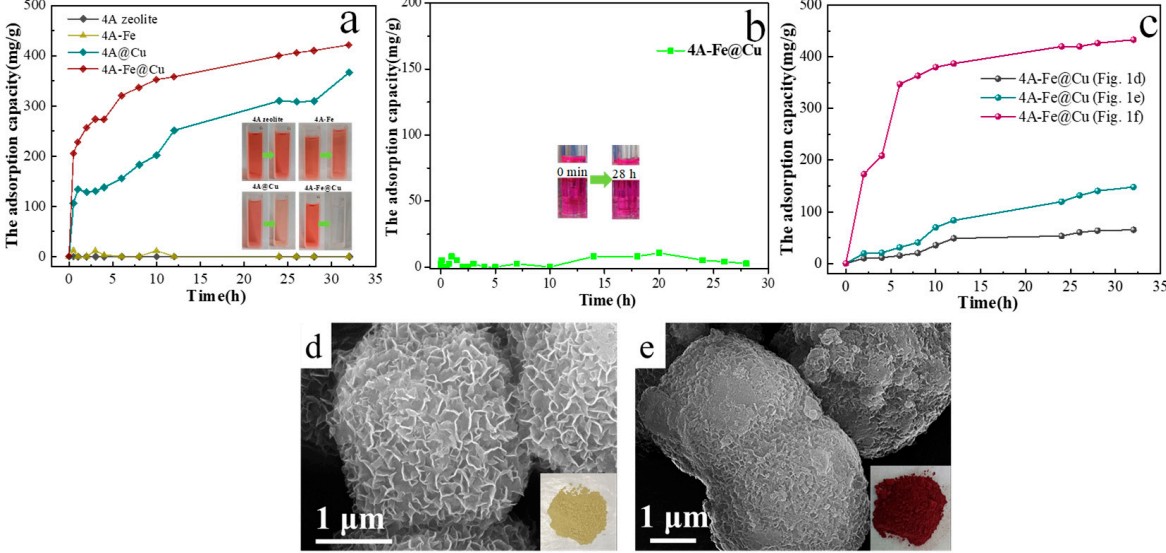

**Figure 3.** (**a**) The CR adsorption capacity of various catalysts, (**b**) the RhB adsorption capacity of 4A-Fe@Cu, (**c**) the CR adsorption capacity of 4A-Fe@Cu with different structures. The insets are the digital photograph of CR and Rhodamine B solution before and after adsorption. SEM images of 4A-Fe@Cu before (**d**) and after (**e**) adsorption. The insets are digital photographs of 4A-Fe@Cu before and after adsorption. (Reaction conditions: 1 g/L of dye solution, 2 g/L catalyst, pH 8).

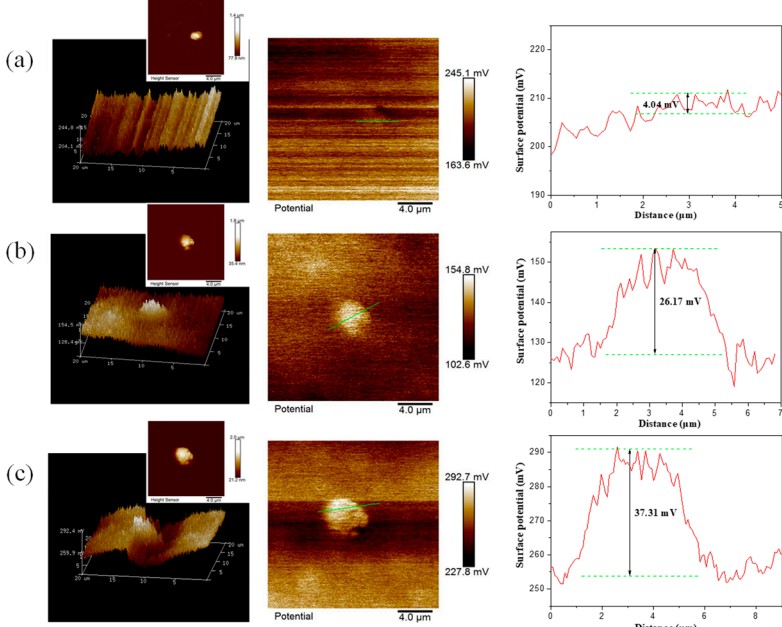

**Figure 4.** The scanning KPFM 3D and 2D images of 4A-Fe (**a**), 4A@Cu (**b**), and 4A-Fe@Cu (**c**) and corresponding surface potential differences along the green line. The insets are the AFM images of various catalysts.

Before investigating the effect of pH on the adsorption performance of 4A-Fe@Cu for CR, the stability of the CR solution was studied, and the results are shown in Figure S2. As the pH value decreased from 10 to 2, the absorption peak of the CR at 500 nm disappeared, and a new absorption peak appeared at 560 nm. The absorption peak became wider, and the color of the CR solution changed from red to blue-purple. This is because the amino groups in Congo red undergo protonation to form an azo protonation structure under acidic conditions and continue to isomerize and transfer into a quinoid structure. This increases the conjugation degree of the molecule, resulting in the maximum absorption wavelength shifting in the direction of the long wave, and the color becomes blue-purple. To avoid the discoloration of CR in experiment, the pH range of the experiment was set to 6–9. The results (Figure 5) show that the adsorption capacity of 4A-Fe@Cu changes slightly with the increased pH.

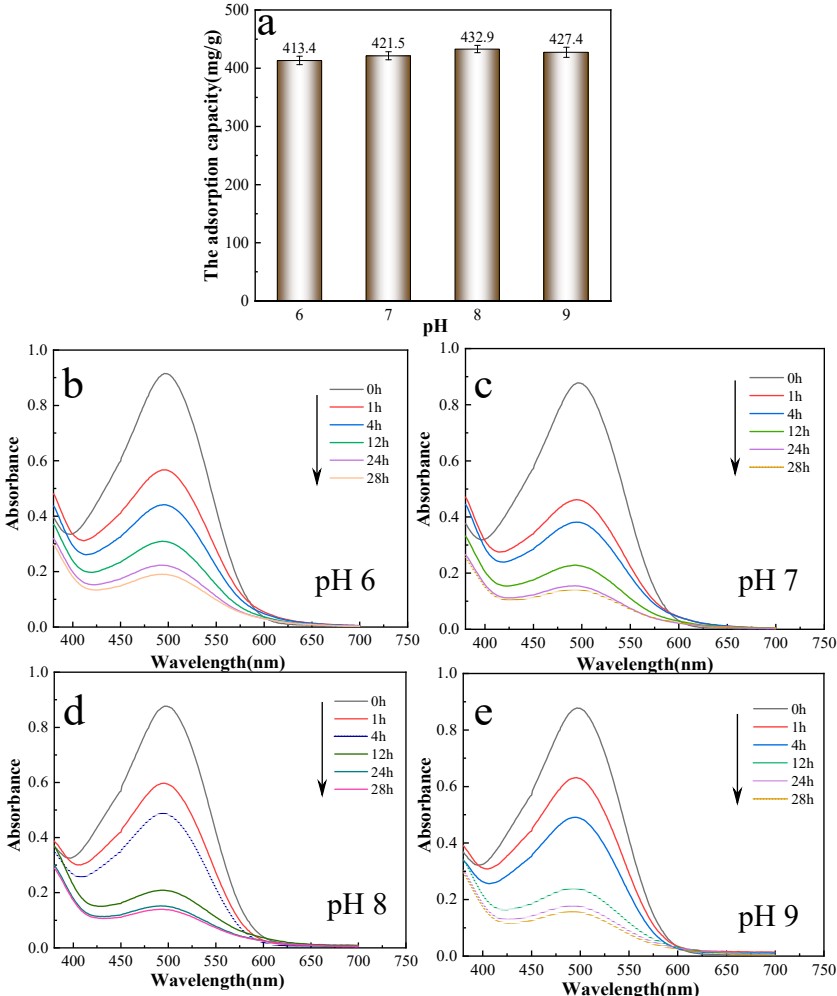

**Figure 5.** The adsorption capacity of 4A-Fe@Cu at different pH (**a**) and corresponding UV-vis spectra at pH 6 (**b**), pH 7 (**c**), pH 8 (**d**) and pH 9 (**e**). Reaction conditions: 1 g/L of CR solution, 2 g/L catalyst.

To investigate the adsorption kinetics of 4A-Fe@Cu, the obtained data were formulated by both the pseudo-first-order and pseudo-second-order models, as shown in Figure 6 and Table 1. The fitting correlation coefficient $R^2$ values of pseudo-second-order kinetics are higher than that of pseudo-first-order kinetics, indicating that the adsorption kinetics of CR on 4A-Fe@Cu are more consistent with the pseudo-second-order model. The parameter calculated $Q_e$ is close to experimental data. This confirms that the adsorption between CR and the catalyst is chemisorption due to electric charge.

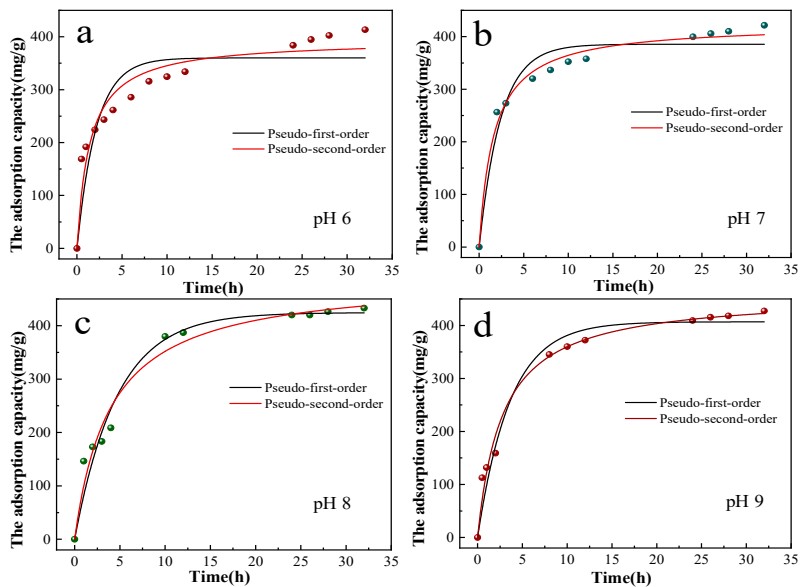

**Figure 6.** Adsorption kinetics of CR on 4A-Fe@Cu at pH 6 (**a**), pH 7 (**b**), pH 8 (**c**) and pH 9 (**d**). Reaction conditions: 1 g/L of CR solution, 2 g/L catalyst.

**Table 1.** Kinetic fitting parameters of CR adsorption.

| | Preudo-First-Order | | | Preudo-Second-Order | | |
|---|---|---|---|---|---|---|
| Parameters | Qe (mg/g) | $k_1$ (h$^{-1}$) | $R^2$ | Qe (mg/g) | $k_2 \times 10^3$ (g/(mg·h)) | $R^2$ |
| pH 6 | 360.0 | 0.468 | 0.825 | 394.4 | 1.782 | 0.925 |
| pH 7 | 385.5 | 0.415 | 0.940 | 424.2 | 1.436 | 0.986 |
| pH 8 | 424.4 | 0.213 | 0.968 | 490.8 | 0.515 | 0.972 |
| pH 9 | 406.9 | 0.276 | 0.968 | 460.0 | 0.789 | 0.986 |

The adsorption isotherm of 4A-Fe@Cu for CR is shown in Figure 7. The regression parameters were calculated by the Langmuir and Freundlich models, and the results are listed in Table 2. The fitting correlation coefficient $R^2$ of the Freundlich model (0.987) is higher than that of the Langmuir model (0.826) (Table 2), indicating that the adsorption process of CR onto the 4A-Fe@Cu surface is multi-molecular layer adsorption. This suggests that the adsorption sites on the catalyst are not uniform.

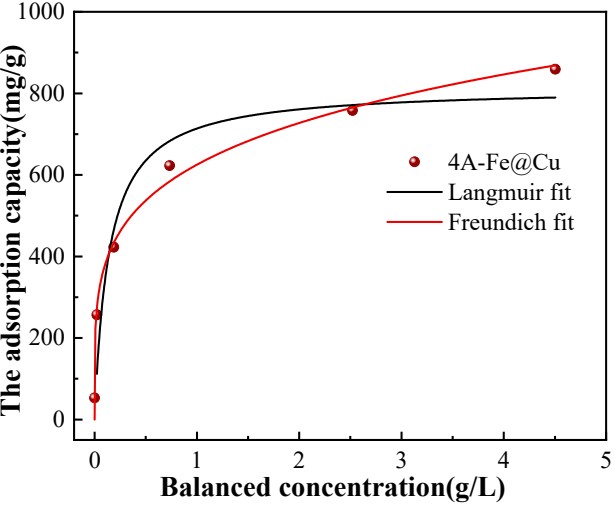

**Figure 7.** Adsorption isotherms of CR onto 4A-Fe@Cu.

**Table 2.** Langmuir and Freundlich isotherm models fitting parameters.

| Langmuir Model | | | Freundlich Model | | |
|---|---|---|---|---|---|
| $Q_m$ (mg/g) | $K_L$ (L/g) | $R^2$ | 1/n | $K_F$ ((mg/g)·(L/g)$^{1/n}$) | $R^2$ |
| 814.5 | 7.086 | 0.826 | 0.219 | 625.071 | 0.987 |

*2.3. Catalytic Performance*

The catalytic degradation of CR dyes by different catalysts is shown in Figure 8. The 4A-Fe catalyst shows catalytic inertia at high pH (pH 8), and the CR can hardly be degraded. Nevertheless, copper-containing catalysts exhibit good catalytic performance at high pH. This confirms that the introduction of the Cu element can broaden the pH range of the Fenton reaction. The 4A-Fe@Cu shows the highest catalytic performance, and the CR degradation ratio reaches 99.2% after 12 h. This indicates that under the same conditions, the 4A-Fe@Cu catalyst can decompose more $H_2O_2$ modules to produce $^{\bullet}OH$ to oxidize CR than 4A@Cu, due to the synergetic effect between iron and copper. The Fenton reaction is essentially a redox or electron transfer reaction. It is well known that the work function represents the electron migration capacity, and the smaller work function value indicates a greater ability of electrons to migrate to the solid surface. Therefore, the work function of catalysts was tested, and the results (Figure 8b) show that the 4A-Fe@Cu possesses the lowest work function value. This suggests that the highest catalytic activity of 4A-Fe@Cu is caused by high electron transfer capability [50–53].

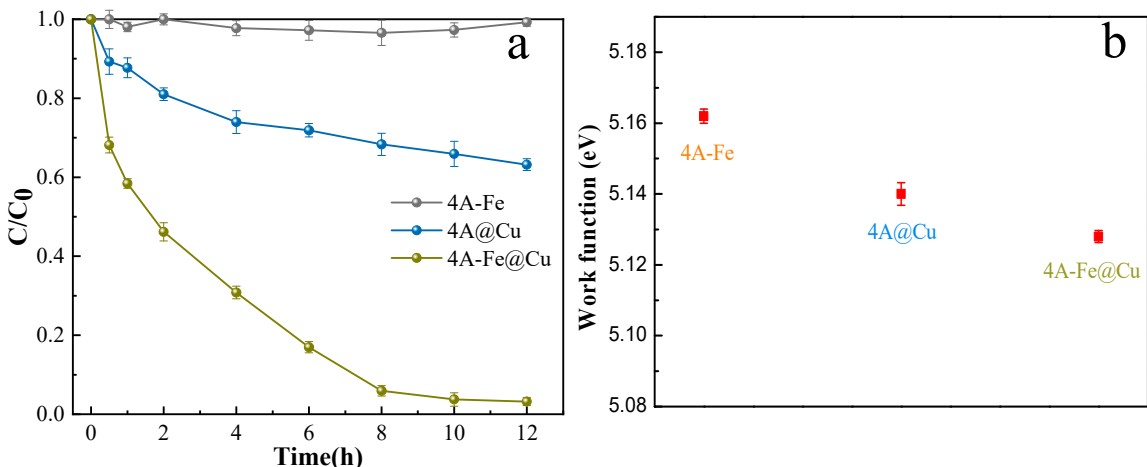

**Figure 8.** (**a**) Degradation of CR dye by different catalysts (Reaction conditions: 1 g/L of CR solution, 7.2 mM $H_2O_2$, 2 g/L catalyst, pH 8). (**b**) The work function of various catalysts.

2.3.1. The Effect of $H_2O_2$ Dosage

The pH of the solution, adding an amount of $H_2O_2$ and catalyst always has a great influence on Fenton reaction efficiency. To further investigate the catalytic performance of 4A-Fe@Cu, the effect of $H_2O_2$ dosage on the degradation of CR by 4A-Fe@Cu was first studied, and the results are shown in Figure 9. The degradation of CR is slow when the mole ratio of $H_2O_2$ dosage to CR is 1:1 ($H_2O_2$ concentration: 1.4 mM). As the mole ratio of $H_2O_2$ dosage to CR increases to 5:1, the degradation of CR is rapidly increased. However, when the dosage of $H_2O_2$ is further increased, the degradation trend of CR does not change much. This is mainly because the excess $H_2O_2$ would consume the $^{\bullet}OH$. The degradation data were fitted by pseudo-first-order and pseudo-second-order kinetic equations, and the calculated parameters are shown in Table 3. In the case of low (1:1) or excessive (60:1, 90:1) $H_2O_2$ content, the reaction is fitted well by pseudo-second-order kinetics, indicating that the reaction rate is controlled by the concentration of CR and $^{\bullet}OH$. When the mole ratio range of $H_2O_2$ to CR is 5:1 to 40:1, the reaction is fitted well by pseudo-first-order kinetics, indicating that the reaction rate is only affected by the concentration of $^{\bullet}OH$.

However, whether the degradation process is fitted well by pseudo-first-order kinetics or pseudo-second-order kinetics, the reaction rate constant increases first and then decreases with the increase of $H_2O_2$ content. When the mole ratio of $H_2O_2$ to CR is 5:1 (7.2 mM), the reaction rate constant reaches maximum.

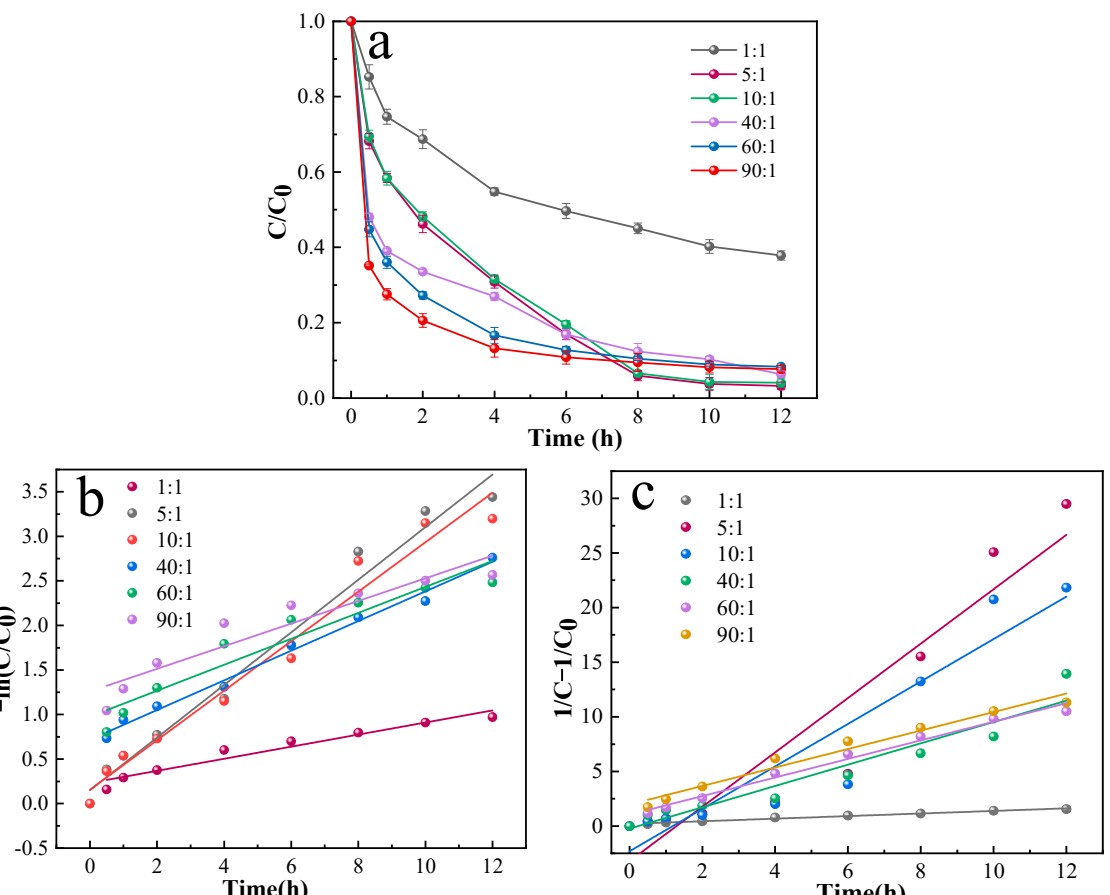

**Figure 9.** The effect of mole ratio of $H_2O_2$ to CR on the catalytic performance of 4A-Fe@Cu (**a**) and corresponding pseudo-first-order (**b**) and pseudo-second-order kinetic (**c**) curves. Reaction conditions: 1 g/L CR (100 mL), 2 g/L catalyst, pH 8.

**Table 3.** Regression parameters of catalytic kinetics of CR by 4A-Fe@Cu with the effect of $H_2O_2$ dosage.

| Parameters | Pseudo-First-Order | | Pseudo-Second-Order | |
|---|---|---|---|---|
| | $k_1$ (h$^{-1}$) | $R^2$ | $k_2$ (L/(g·h)) | $R^2$ |
| 1:1 | 0.0678 | 0.942 | 0.118 | 0.982 |
| 5:1 | 0.295 | 0.977 | 2.490 | 0.884 |
| 10:1 | 0.278 | 0.968 | 1.943 | 0.884 |
| 40:1 | 0.166 | 0.989 | 0.976 | 0.914 |
| 60:1 | 0.146 | 0.904 | 0.850 | 0.985 |
| 90:1 | 0.127 | 0.873 | 0.846 | 0.970 |

### 2.3.2. The Effect of Catalyst Dosage

The influence of catalyst dosage on the degradation of CR by 4A-Fe@Cu is shown in Figure 10. The degradation of CR is accelerated as the mass ratio of 4A-Fe@Cu to CR increases from 1:2 to 2:1. When the catalyst dosage further increases, the degradation curve of CR tends to be gentle. The regression analysis parameters are listed in Table 4. Similar to the influence of $H_2O_2$ amount on the catalytic process (Figure 9), the degradation process of CR affected by catalyst concentration also follows pseudo-second-order kinetic when

the amount of catalyst is (1:2) small or large (4:1). When the mass ratio of catalyst to CR is between 1:1 and 2:1, the degradation of CR conforms to pseudo-first-order.

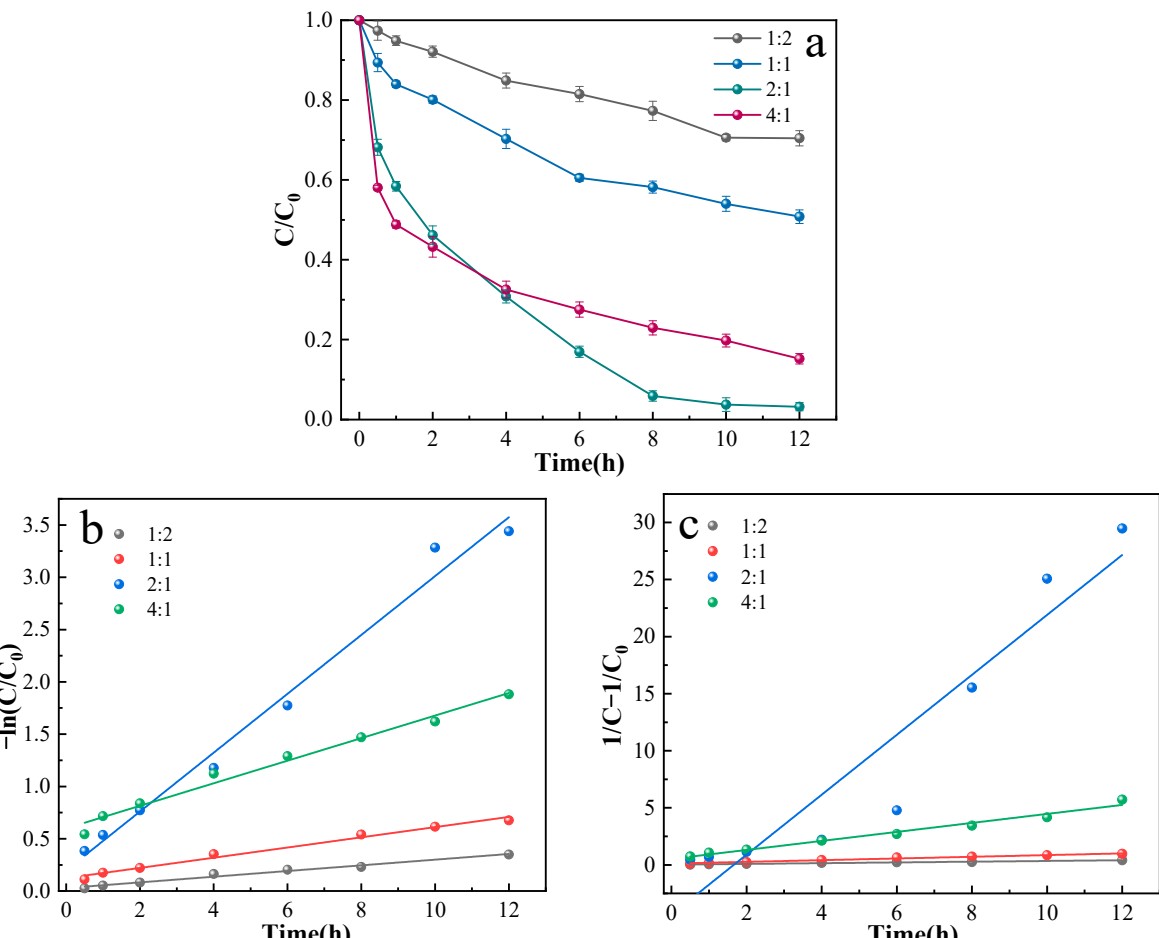

**Figure 10.** The influence of catalyst dosage on the catalytic performance of 4A-Fe@Cu (**a**) and corresponding pseudo-first-order (**b**) and pseudo-second-order kinetic (**c**) curves. Reaction conditions: 1 g/L CR (100 mL), 7.2 mM $H_2O_2$, pH 8.

**Table 4.** Regression parameters of catalytic kinetics of CR by 4A-Fe@Cu with the effect of catalyst dosage.

| | Pseudo-First-Order | | Pseudo-Second-Order | |
|---|---|---|---|---|
| **Parameters** | $k_1$ (h$^{-1}$) | $R^2$ | $k_2$ (L/(g·h)) | $R^2$ |
| 1:2 | 0.028 | 0.979 | 0.032 | 0.985 |
| 1:1 | 0.048 | 0.984 | 0.074 | 0.982 |
| 2:1 | 0.281 | 0.984 | 2.626 | 0.887 |
| 4:1 | 0.109 | 0.980 | 0.394 | 0.975 |

### 2.3.3. The Effect of pH

The pH is an important factor that affects the Fenton reaction. The influence of the initial pH of CR solution on the catalytic performance of 4A-Fe@Cu is shown in Figure 11, and the regression analysis parameters are listed in Table 5. With the increase of pH, the catalytic efficiency decreases obviously. However, it is worth mentioning that the 4A-Fe@Cu catalyst can still effectively degrade CR at weakly alkaline conditions (pH 8). Compared with the traditional Fenton catalyst, which is only suitable for acidic conditions, this catalyst greatly broadens the pH conditions of the Fenton reaction. According to the analysis in Table 5, the catalytic reaction follows a pseudo-first-order dynamic when the pH of the initial solution changes.

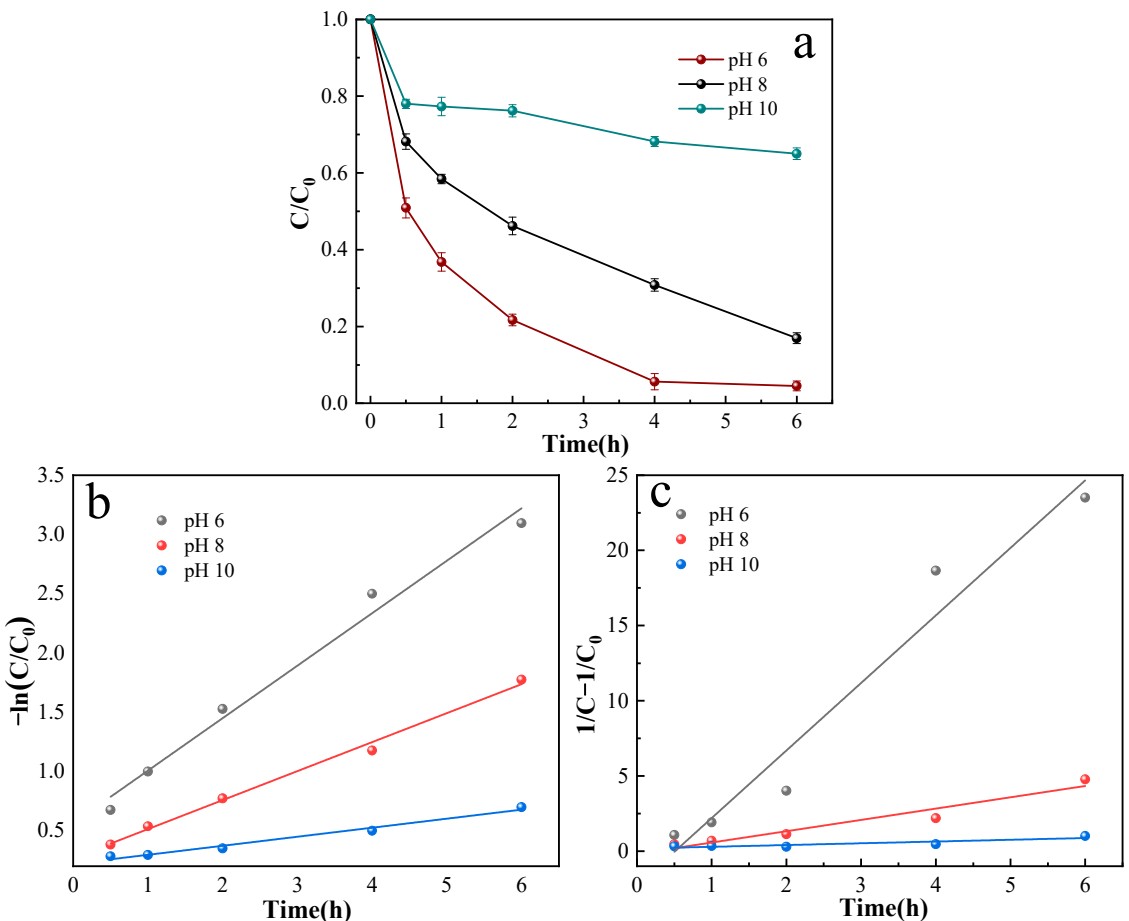

**Figure 11.** The influence of initial pH of CR solution on the catalytic performance of 4A-Fe@Cu (**a**) and corresponding pseudo-first-order (**b**) and pseudo-second-order kinetic (**c**) curves. Reaction conditions: 1 g/L CR (100 mL), 7.2 mM $H_2O_2$, 2 g/L catalyst.

**Table 5.** Regression parameters of catalytic kinetics of CR by 4A-Fe@Cu with the effect of pH.

| Parameters | Pseudo-First-Order | | Pseudo-Second-Order | |
|---|---|---|---|---|
| | $k_1$ (h$^{-1}$) | $R^2$ | $k_2$ (L/(g·h)) | $R^2$ |
| pH 6 | 0.443 | 0.980 | 4.491 | 0.943 |
| pH 8 | 0.245 | 0.992 | 0.752 | 0.923 |
| pH 10 | 0.076 | 0.976 | 0.117 | 0.724 |

### 2.4. The Effect of Adsorption on Catalysis

In order to explore the influence of CR adsorption of 4A-Fe@Cu in the catalytic process, two groups of experiments were carried out. One is the degradation of CR by a fresh catalyst, and the other is the degradation of CR by a fully adsorbed catalyst. It can be seen from Figure 12 that the removal rate (adsorption) is approximately 72.1% until 32 h and does not increase with time after this point, indicating the adsorption saturation of 4A-Fe@Cu. Then, the $H_2O_2$ was added, and the catalysis degradation of CR was started with this fully adsorbed 4A-Fe@Cu, and the removal rate (catalysis) of CR in solution was calculated from this point. It is obviously found that the trends and results of the catalytic curves of fully adsorbed 4A-Fe@Cu and fresh 4A-Fe@Cu are almost identical. This suggests that the adsorption property of 4A-Fe@Cu has no negative effect on its catalytic performance; that is to say, the adsorption of CR does not inactivate the catalytic site on 4A-Fe@Cu.



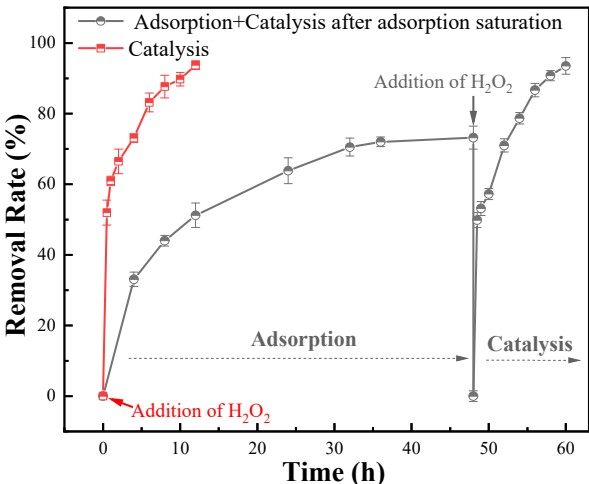

**Figure 12.** The influence of catalyst adsorption process on catalytic performance. Reaction conditions: 1 g/L CR (100 mL), 7.2 mM $H_2O_2$, 2 g/L catalyst, pH 8.

To further investigate the relationship between adsorption and catalytic performance of the 4A-Fe@Cu, the catalysis of CR by 4A-Fe@Cu with different structures (different adsorption performance) was conducted, and the results are shown in Figure 13a. The CR degradation performance of 4A-Fe@Cu is enhanced with the increase of nanosheet structure. There are two main reasons for this; one is that the high specific surface area of the catalyst with an open network structure increases the catalysis sites, and the other is that the high adsorption capacity reduces diffusion mass transfer resistance and improves the accessibility of CR to the catalyst. In addition, the degradation of RhB, which cannot be adsorbed by 4A-Fe@Cu, was run. Figure 13b indicates that the degradation ratio of RhB is much lower than that of CR (Figure 13a) by 4A-Fe@Cu under the same conditions. The good adsorption capacity accompanied by high decolorization of the CR again demonstrates that the adsorption can enhance the degradation of CR. The possible mechanisms for the removal of CR by 4A-Fe@Cu catalysts are proposed and shown in Figure 14. The $Fe^{3+}$ reacts with $H_2O_2$ to form $Fe^{2+}$ and $HO_2^{\bullet}/O_2^{\bullet-}$, and then $Fe^{2+}$ reacts rapidly with $H_2O_2$ to form $^{\bullet}OH$. Meanwhile, $Cu^{2+}$ on 4A-Fe@Cu can be reduced to $Cu^+$ by $HO_2^{\bullet}/O_2^{\bullet-}$ or $H_2O_2$, and then $Cu^+$ can react with $H_2O_2$ to produce $^{\bullet}OH$. In addition, $Cu^+$ can promote the transformation of $Fe^{3+}$ to $Fe^{2+}$ due to thermodynamically favorable [54]. In addition, the comparison between the adsorption and catalysis of CR (Figure S3) verifies the functionality of the catalysis process. As compared to adsorption, catalysis has high de-coloration efficiency.

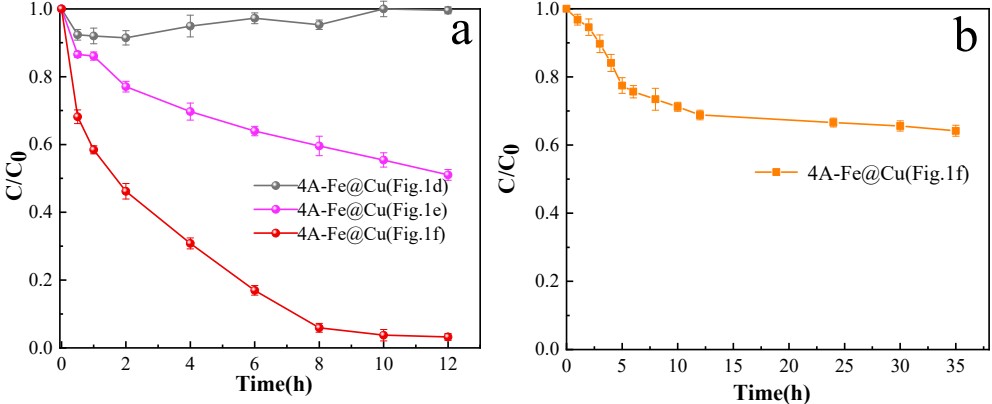

**Figure 13.** (**a**) Degradation of CR by 4A-Fe@Cu with different structures, (**b**) degradation of RhB by 4A-Fe@Cu (Figure 1f). Reaction conditions: 1 g/L RhB or CR (100 mL), 7.2 mM $H_2O_2$, 2 g/L catalyst, pH 8.

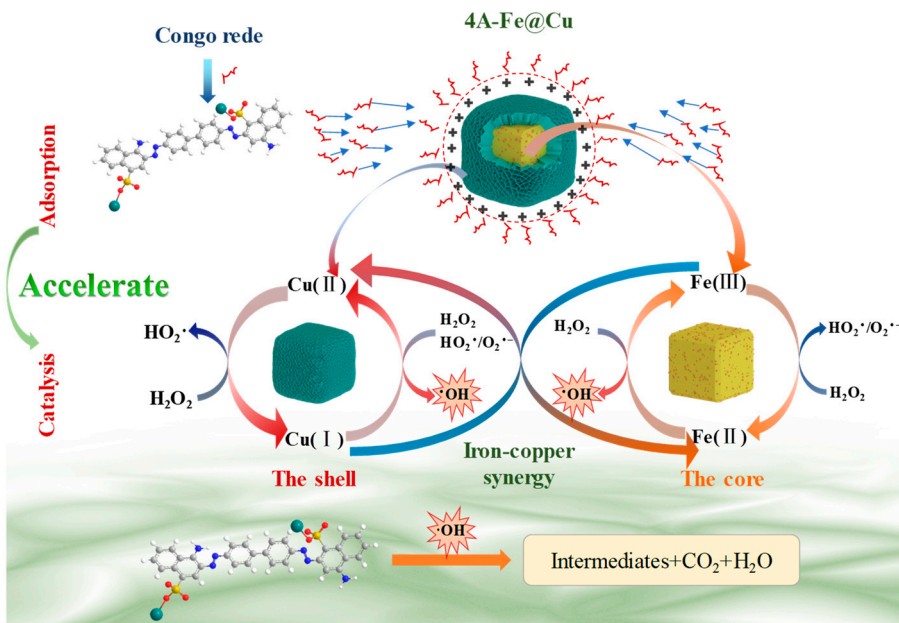

**Figure 14.** Proposed mechanism for the enhanced adsorption and degradation reaction of CR.

The morphologies of the above 4A-Fe@Cu after adsorption and/or catalysis were characterized by SEM and TEM. Figure 15a shows that the nanosheets, after adsorption for 12 h, are wrapped with a thick layer of CR molecules, and the porous structure becomes less obvious. However, the morphology structure of 4A-Fe@Cu hardly changes after 12 h of catalysis (Figure 15b), compared with the fresh 4A-Fe@Cu (Figure 1f). For a more intuitive analysis, the TEM images of 4A-Fe@Cu after adsorption and catalysis for 12 h are shown in Figure 15c–f. It can be seen that the nanosheets on the surface of 4A-Fe@Cu adsorbed CR are not stretched but clumped together. Nevertheless, the nanosheets on 4A-Fe@Cu after catalysis stretches just like the fresh catalyst.

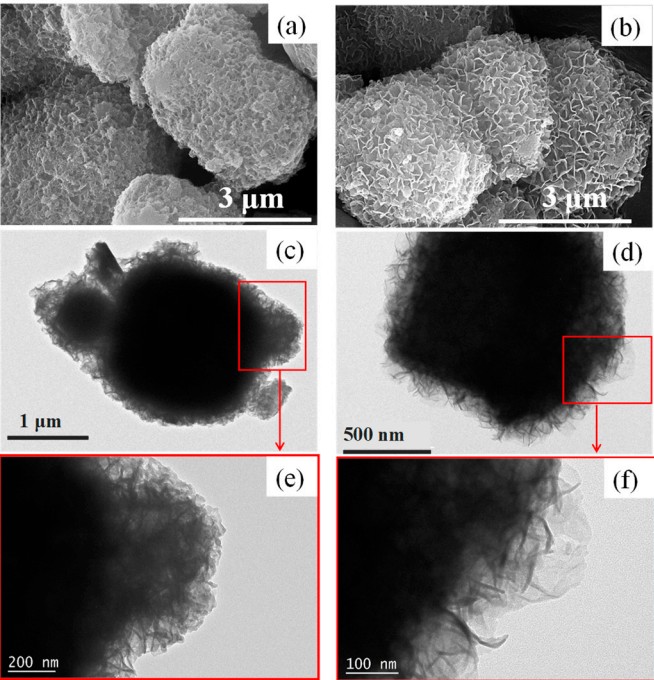

**Figure 15.** SEM and TEM images of 4A-Fe@Cu after 12 h of adsorption and catalysis ((**a,c,e**): adsorption; (**b,d,f**): catalysis).

### 2.5. Comparison with Other Fe/Cu Bimetallic Catalysts

A comparison was made to obtain an overview of the catalytic performance of 4A-Fe@Cu for CR degradation, and the results are shown in Table 6. The 4A-Fe@Cu exhibits superior adsorption and catalytic performance for CR, which indicates that the fabrication of a catalyst with both adsorption and catalytic capacities is an efficient strategy to enhance the removal of CR.

**Table 6.** Comparison of the catalytic performance with other Fe/Cu bimetallic catalysts.

| Catalyst | Pollutant | Pollutant Concentration (g/L) | Catalyst Concentration (g/L) | $H_2O_2$ (mM) | pH | T (°C) | Degradation Ratio % | Adsorption Capacity (mg/g) | Ref. |
|---|---|---|---|---|---|---|---|---|---|
| (Mg,Cu,Ni)(Fe,Al)$_2$O$_4$ | RhB | 0.01 | 1.0 | 24.5 | / | 45 | 97.3 | / | [55] |
| Fe$_3$O$_4$/C/Cu | MB | 0.1 | 0.5 | 163.7 | 6.9 | 35 | 85.0 | / | [56] |
| 5Fe$_{2.5}$Cu-Al$_2$O$_3$ | NB | 0.1 | 1.0 | 29.4 | 3.0 | 50 | 94.0 | / | [57] |
| CuFe-MC | BPA | 0.1 | 0.3 | 30.0 | 5.0 | 25 | 58.0 | 132.0 | [58] |
| Fe Cu | Orange II | 0.02 | 0.5 | 10.0 | 6 | 25 | 90.0 | 32.0 | [59] |
| Cu/Fe$_3$O$_4$@CRC | MB | 0.04 | 0.2 | 4.0 | 6 | 25 | 97.5 | 240.3 | [60] |
| (Fe,Cu)S/CuFe$_2$O$_4$ | EE2 | 0.005 | 0.5 | 20.0 | 6.5 | 25 | 73.0 | 5.7 | [61] |
| 4A-Fe@Cu | CR | 1.0 | 2.0 | 7.2 | 8 | 25 | 99.2 | 432.9 | This work |
| 4A-Fe@Cu | CR | 1.0 | 2.0 | 14.4 | 8 | 25 | 99.0 | 432.9 | This work |
| 4A-Fe@Cu | CR | 1.0 | 2.0 | 57.6 | 8 | 25 | 93.7 | 432.9 | This work |

## 3. Materials and Methods

### 3.1. Materials

4A zeolite was purchased from Shanghai Jiu Zhou Chemicals Co., Ltd. (Shanghai, China). Ferric chloride (FeCl$_3$) and copper chloride (CuCl$_2$) were obtained from Tianjin Komiou Chemical Reagent Co., Ltd. (Tianjin, China). Congo red (CR) and hydrogen peroxide (H$_2$O$_2$, 30 wt.%) were obtained from Aladdin Chemical Reagent (Shanghai, China). Hydrochloric acid (HCl) and concentrated sulfuric acid (H$_2$SO$_4$, 98 wt.%) were purchased from Sinopharm Chemical Reagent Co., Ltd. Sodium hydroxide (NaOH) was purchased from Tianjin Zhiyuan Chemical Reagent Co., Ltd. (Tianjin, China). All chemicals were of analytical grade and used without further purification. Deionized water was used throughout the experiment.

### 3.2. Fabrication of 4A-Fe@Cu Catalysts

The preparation of 4A-Fe@Cu is shown in Figure 16. First, 500 mL of FeCl$_3$ solution with a concentration of 2.0 g/L was prepared, and the pH of the solution was adjusted to 1.67 by HCl (1.0 mol/L). Then, 10 g of 4A zeolite was added into the prepared solution and stirred for 3.5 h; yellow 4A-Fe could be obtained through centrifugal washing and freeze-drying at −50 °C for 24 h.

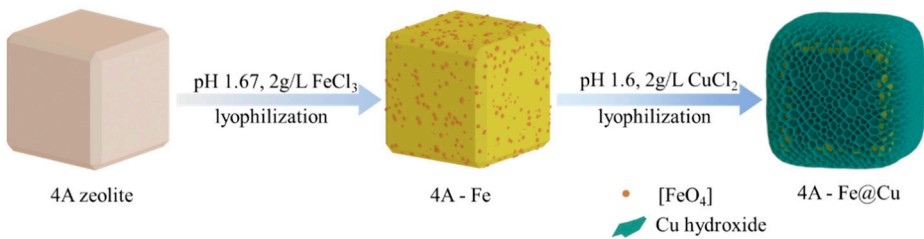

**Figure 16.** The schematic diagram of catalyst preparation.

Second, 200 mL of CuCl$_2$ solution with a Cu$^{2+}$ concentration of 2.0 g/L was prepared, and the pH was adjusted to 1.6 by diluting H$_2$SO$_4$ (1.0 mol/L). Then, as prepared, 4 g of the 4A-Fe was added into the CuCl$_2$ solution and stirred for 5 h. After centrifugal washing three times and freeze-drying at −50 °C for 24 h, the yellow-green 4A-Fe@Cu catalyst was obtained. The effect of the pH of the CuCl$_2$ solution on the morphology of the 4A-Fe@Cu catalyst was investigated. In addition, as a comparative experiment, the 4A@Cu catalyst was the prepared basis on 4A zeolite via the same method.

### 3.3. Characterization

The size and morphology of the prepared 4A-Fe@Cu catalyst were studied by scanning electron microscopy (SEM, Nicolet iS20, Thermo Scientific, Waltham, MA, USA) and transmission electron microscope (TEM, Tecnai G2F 20, FEI). Energy dispersive spectroscopy (EDS, JEOL-2100F) was used to analyze the surface element distribution of the catalyst. The element composition of the catalyst was detected by Fourier transform infrared spectroscopy (FT-IR, Nicolet iS20, Thermo Scientific, Waltham, MA, USA) and X-ray photoelectron spectroscopy (XPS, K-Aepna, Thermo Scientific, Waltham, MA, USA). The physical structure of catalysts was checked using X-ray diffraction (XRD, D8 Discover, Bruker, Karlsruhe, Germany) at a scanning speed of $10°/\text{min}$. The concentration of CR solution at different times was tested with an ultraviolet-visible spectrophotometer (UV-vis, Mapada UV-1100). The surface morphology and potential of the catalyst were measured by atomic force microscopy (AFM) combined with kelvin probe force microscopy (KPFM, Bruker Dimension Icon, Karlsruhe, Germany). Amplitude modulation KPFM was used to obtain a high signal-to-noise ratio as opposed to that of frequency modulation. All KPFM measurements were performed in dual-pass mode to eliminate the topography effect completely. The work function values of catalysts were calculated by measuring the potential of the Au sample, the work function of which was generally identified as 5.2 eV.

### 3.4. Adsorption Test

#### 3.4.1. Adsorption Kinetics

First, 100 mL of CR solution with a concentration of 1.0 g/L was prepared, and the pH was adjusted to 7–10 by using HCl or NaOH. Subsequently, 0.2 g of catalyst was added to the solution and placed in a shaker at 120 rpm for adsorption experiments. At different time intervals, 1 mL solution was pipetted and filtrated with a 0.45 μm filter. The CR concentration was analyzed by UV-vis at 496 nm. As a comparison, the adsorption of cationic dyes (Rhodamine B, RhB) by 4A-Fe@Cu was conducted at the same above conditions. The adsorption capacity was calculated according to the following Equation (1):

$$Q = \frac{(C_0 - C_t)V}{W} \tag{1}$$

where Q represents the adsorption capacity (mg/g) of CR or RhB, and $C_0$ and $C_t$ represent the CR or RhB concentration at an initial and arbitrary time (mg/L), respectively. V represents the volume (L) of the solution, and W represents the mass (g) of the adsorbent.

The effect of the initial pH of the solution on the adsorption performance of 4A-Fe@Cu for CR was investigated, and the process was adopted by pseudo-first-order and pseudo-second-order equations.

The pseudo-first-order model:

$$\log(Q_e - Q_t) = \log Q_e - \frac{k_1}{2.303}t \tag{2}$$

The pseudo-second-order model:

$$\frac{t}{Q_t} = \frac{1}{k_2 \cdot Q_e^2} + \frac{1}{Q_e}t \tag{3}$$

where $Q_e$ (mg/g) and $Q_t$ (mg/g) are the adsorption capacity at equilibrium and at any time t, respectively. $k_1$ $(\text{h}^{-1})$ and $k_2$ $(\text{g}/(\text{mg}\cdot\text{h}))$ are the rate constants of the kinetic models.

#### 3.4.2. Adsorption Isothermal

The adsorption isotherm was measured for CR adsorption on 4A-Fe@Cu by batch experiments. Typically, 100 mL of CR solution with a concentration of 0.1, 0.5, 1.0, 2.0, 4.0, and 6.0 g/L were prepared, respectively, and placed into a 150 mL conical flask. Then 0.2 g of 4A-Fe@Cu was added, and the conical flask was placed in a shaking bath. Langmuir isotherm model and Freundlich isotherm model were adopted.

The Langmuir model:

$$\frac{C_e}{Q_e} = \frac{C_e}{Q_m} + \frac{1}{Q_m K_L} \tag{4}$$

where $Q_e$ is the equilibrium adsorption capacity (mg/g), $C_e$ is the equilibrium concentration (g/L), $Q_m$ is the maximum adsorption capacity using a curve fitting (mg/g), and $K_L$ is the Langmuir equilibrium constant (L/g).

The Freundlich isotherm model:

$$Q_e = K_F C_e^{\frac{1}{n}} \tag{5}$$

where $Q_e$ and $C_e$ have the same definitions as before, $K_F$ ($(mg/g)\cdot(L/g)^{1/n}$) and n are the Freundlich constants.

### 3.5. Catalysis Test

The catalytic performance of the as-prepared catalyst was also investigated by batch experiments. In greater detail, 100 mL CR solution with a concentration of 1.0 g/L was placed into a 150 mL conical flask. Then 0.2 g catalyst was added and shaken. The reaction was triggered by the addition of $H_2O_2$. At different time intervals, 1 mL solution was pipetted and filtrated with a 0.45 μm filter. The CR concentration was analyzed by UV-vis. The effects of $H_2O_2$ dosage, catalyst dosage, and pH of solution on catalytic performance were studied, and the catalytic processes were adopted by first-order and second-order kinetics (Equations (6) and (7)).

$$-\ln\frac{C}{C_0} = k_1 t \tag{6}$$

$$\frac{1}{C} - \frac{1}{C_0} = k_2 t \tag{7}$$

where C and $C_0$ represent the CR concentration at an initial and arbitrary time (g/L) in the degradation process, respectively. $k_1$ ($h^{-1}$) and $k_2$ (L/(g·h)) are the rate constants of the kinetic models.

### 4. Conclusions

In this work, a core-shell hierarchical 4A-Fe@Cu bimetallic Fenton catalyst was prepared by a simple and effective method. The 4A-Fe@Cu exhibited high adsorption capacity and excellent catalytic performance for anionic dye CR. The good adsorption property of 4A-Fe@Cu for CR significantly enhanced the catalysis process. In addition, the 4A-Fe@Cu bimetallic catalyst exhibited higher catalytic activity than monometallic 4A@Cu and/or 4A-Fe catalysts due to lower work function value. This strategy provided guidance to the design of high-performance Fenton-like catalysts with both adsorption and catalysis properties for dye wastewater treatment.

**Supplementary Materials:** The following supporting information can be downloaded at: https://www.mdpi.com/article/10.3390/catal12111363/s1; Figure S1: The standard curve of Congo red; Figure S2: The effect of pH on the absorbance of Congo red solution; Figure S3: Adsorption and catalysis comparison; Figure S4: Iron (a) and Copper (b) leaching in CR degradation of 4A-Fe@Cu. Reaction conditions: pH 8, 2 g/L catalysts, 1 g/L CR solution, 7.2 mM $H_2O_2$.

**Author Contributions:** Conceptualization, H.C., S.W. and L.H.; methodology, H.C., L.Z. and J.P.; software, J.H. and W.R.; validation, L.Z. and J.P.; formal analysis, J.H.; investigation, H.C., L.Z. and J.H.; resources, J.L.; data curation, H.C. and L.H.; writing—original draft preparation, H.C.; writing—review and editing, S.W. and L.H.; visualization, W.R.; supervision, J.L.; project administration, L.H. and J.L.; funding acquisition, S.W. and J.L. All authors have read and agreed to the published version of the manuscript.

**Funding:** The authors gratefully acknowledge the financial support from the Shandong Provincial Natural Science Foundation (ZR2020ME051), the Provincial Natural Science Foundation of Hunan

(2022JJ40041), the Innovation Research Fund of Zhaoyuan Institute of Industrial Technology (Grant No. 220192).

**Data Availability Statement:** Data are available from the authors.

**Conflicts of Interest:** The authors declare no conflict of interest.

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
