# Peer review of "Core-Shell Hierarchical Fe/Cu Bimetallic Fenton Catalyst with Improved Adsorption and Catalytic Performance for Congo Red Degradation"

_catalysts, doi:10.3390/catal12111363_

Round 1

Reviewer 1 Report

(1)      The writing style of H2O2 needs to be uniform throughout manuscript.

(2)      “Fe/Cu bimetallic catalyst” other than “bimetallic catalyst” should be presented in keywords.

(3)      Some relevant references for emerging Fenton-like system for organic pollutants removal, such as 10.1016/j.cej.2020.128176, 10.1016/j.cclet.2021.10.087, 10.1016/j.cej.2022.138588, 10.1016/j.seppur.2022.120716 should be cited in introduction or any other appropriate places.

(4)      The writing style of radicals should be corrected according to the following reference: “Koppenol, W. (2000). "Names for inorganic radicals (IUPAC Recommendations 2000)." Pure and Applied Chemistry 72(3): 437-446.”

(5)      Figure 2 and Figure 3 should be merged. And the number of figures is too plenty, pls merge them.

(6)      Pls add error bars in Figure 10a, 11a, 12a, 13a, 14, 15.

(7)      Pls provide details for CR analyze.

(8)      What about the Cu and Fe leaching during CR degradation?

Reviewer 2 Report

This paper reports a core-shell hierarchical Fe/Cu bimetallic Fenton catalyst with improved adsorption and catalytic performance for Congo red degradation. The authors demonstrated the good adsorption and catalytic degradation properties of 4A-Fe@Cu catalyst for Congo red in detail. However, following issues should be addressed before the publication of this work.

1.        The authors claim that the relationship between adsorption and catalysis has been studied intensively in Introduction section. However, the adsorption property of 4A-Fe@Cu has no negative effect on its catalytic performance (Figure 14). Hence, the result is confusing.

2.        The major irregularity lies in molecular formula (e.g. H2O2 in Abstract) and numerical value (e.g. 137, 116.06 and 132.15 mg/g)

3.        The authors should measure the thickness of nanosheets carefully. It is obvious that the thickness of nanosheets is more than 100 nm form TEM images (Figure 2a-b).

4.        The adsorption capacity of the 4A@Cu catalyst for CR is missing when compared to the 4A-Fe@Cu catalyst.

Reviewer 3 Report

Title: Core-shell hierarchical Fe/Cu bimetallic Fenton catalyst with improved adsorption and catalytic performance for Congo red degradation

Authors: Haimei Chen, Shaofei Wang, Lilan Huang, Leitao Zhang, Jin Han, Wanzheng Ren, Jian Pan, Jiao Li

In this paper, Chen and coworkers report the preparation and characterization of a 4A-Fe@Cu bimetallic Fenton catalysts with a three-dimensional core-shell structure by a simple, template-free and surfactant-free methodology The as-prepared catalysts were used in the adsorption and degradation of Congo red (CR).

The structure of the article fulfills the structure of a research article. The manuscript is well organized, very interesting for the increasing community working on the development of new catalysts with superior performance.

This is a good paper, easy to read and I recommend the publication after minor revision.

In my opinion, there is only few points that should be considered for optimizing the manuscript as follows:

  1. Page 2, line 63-64: the sentence should be rephrased.
  2. Almost all the figures have a very low resolution – please improve the quality of the next figures: Figure 2, Figure 3, Figure 4, Figure 5, Figure 7, Figure 8, Figure 10, Figure 11, Figure 12, Figure 13;
  3. Please write the concentration either in mg/mL or g/L;
  4. Page 15, line 449:  “Detailed” not detailedly.
